# Laparoscopy-assisted percutaneous correction of abdominal wall defects in the umbilical region in a cadaveric model of bovine fetus

**Carla Rozilene Guimarães Silva**[1‡]\*, **Thiago da Silva Cardoso**[1©], **Késia Bandeira da Silva**[1©], **Heytor Jales Gurgel**[1©], **João Pedro Monteiro Barroso**[1©], **Luiz Henrique Vilela Araújo**[1©], **Luis Enrique Soza Altamirano**[1©], **Loise Araújo de Sousa**[1©], **Luiza Paula Araújo Alcântara**[1©], **Marcos Emanoel Martins Ferreira**[1©], **Lucas Santos Carvalho**[1©], **José Leandro da Silva Gonçalves**[1©], **Jhoisse Hamar Guimarães Rodrigues**[1©], **Francisco Décio de Oliveira Monteiro**[1‡], **Rinaldo Batista Viana**[2‡], **Pedro Paulo Maia Teixeira**[1‡]

1 Institute of Veterinary Medicine, Federal University of Pará (UFPA), Castanhal, Pará, Brazil, 2 Institute of Animal Health and Production, Federal Rural University of Amazonia (ISPA/UFRA), Belém-Pará, Brazil

© These authors contributed equally to this work.
‡ These authors also contributed equally to this work
\* carlarozilene@hotmail.com

**Data Availability Statement:** All relevant data are within the paper.

## Abstract

Abdominal wall defects in calves are commonly diagnosed and treated via laparotomy. This technique has witnessed several advancements in the management of these disorders. This study aimed to create a study model and evaluate the feasibility of video-assisted percutaneous correction of abdominal wall defects in bovine fetuses (corpses) compared with the conventional technique. Sixteen bovine fetuses from pregnant cows slaughtered in slaughterhouses were included in this study. The fetuses were categorized into the control group (CG, n = 8), which was subjected to umbilical abdominorrhaphy via laparotomy, and the video-surgical group (VG, n = 8), which received video-assisted percutaneous sutures with two lateral accesses on the right flank. An abdominal wall defect was created in the VG group to generate a study model, which was corrected using the laparoscopic technique. The procedures were performed in two steps. The first step consisted of creating an abdominal wall defect in the umbilical region by laparoscopic approach in an iatrogenic manner (Step 1: E1). The second stage consisted of conventional abdominorrhaphy of the umbilical region wall defect in the CG group and video-assisted percutaneous suturing of the edges of the iatrogenic abdominal wall defect in the VG group, until reversal of the laparoscopic accesses (Step 2: E2). Step 1 showed no statistically significant difference between the two groups. However, a significant statistical difference ($p < 0.0001$) was observed between the two groups in step 2. The surgical time of step 2 was longer in the CG group (33.10 ± 0.43 minutes) than that in the VG group (10.13 ± 0.68 minutes, $p < 0.0001$), and the total surgical time was also longer in the CG group (38.48 ± 0.35 minutes) than that in the VG group (15.86 ± 0.67 minutes). The proposed laparoscopic technique allowed the creation of a study model for video-assisted percutaneous suturing with two portals and reduced the

**Funding:** The authors would like to thank CAPES (Coordenação de Aperfeiçoamento de Pessoal de Nível Superior - "Higher Education Personnel Improvement Coordination") and PROPESP (Pró-Reitoria e Pesquisa e Pós Graduação - "Dean of Research and Graduate Studies")/UFPA (Universidade Federal do Pará - "Federal University of Pará) for financially supporting this study. The funders had no role in study design, data collection and analysis, decision to publish, or preparation of the manuscript.

**Competing interests:** The authors have declared that no competing interests exist.

surgical time compared with the conventional technique. However, this method needs to be studied further in live animals.

## Introduction

Breeds with dairy aptitude are more affected by umbilical hernias than beef breeds, and the problem is more common in females [1]. Abdominal wall defects in calves can often occur in production systems that do not perform preventive genetic selection and sporadically in other owners who are attentive to hereditary or congenital problems [2]. Umbilical hernias in calves may be noninfectious in origin, but they can also be infectious or incisional owing to technical failure of umbilical surgeries, errors in suture patterns, and concomitant infections of the umbilical remnants [3,4].

Hernia comprises three parts, namely, hernial ring, sac, and contents. An umbilical hernia usually occurs in calves because of the failure to close the umbilical cord. This failure results in the projection of the abdominal contents into the subcutaneous tissue, which causes a protrusion of the peritoneum and enlargement of the umbilical region [5].

Clinical signs, palpation, needle impression, and auscultation as well as rectal temperature, heart rate, and respiratory rate measurement can aid in the diagnosis [6,7]. Palpation can confirm the diagnosis of umbilical hernia, or even other hernias, as it reduces the hernial content to the abdominal cavity and differentiates it from other conditions, such as abscesses, edema, and idiopathic protrusions. When punctured, the intestinal content may be present in the obtained sample [6].

Ultrasound shows satisfactory results as a complementary diagnostic method, and when complemented by videolaparoscopy, it can aid in exploring the umbilical region, identifying adhesions, and expanding the field of view [8]. Videolaparoscopy is used for the intra-abdominal diagnosis of umbilical remnants, including repair of umbilical hernias in other species. This method exhibits promising results as compared with conventional methods via laparotomy [9–11].

Laparoscopic techniques need to be practiced in study models so that the surgeon acquires the skills to perform the procedure in live animals [9,10]. However, there are no reports on any experimental training model for creating an abdominal wall defect via the videosurgical approach. Also, there is no description of the percutaneous correction technique assisted by abdominal videolaparoscopy.

Hence, this study aimed to create a study model and evaluate the viability of video-assisted percutaneous suturing of the abdominal wall technique in bovine fetuses (corpses) as compared with the conventional technique.

## Materials and methods

This study was approved by the Ethics Committee for Research with Animals and Experimentation of the Federal University of Pará (protocol N° 4848261017).

### Animals

A total of 16 bovine fetuses were included in the study, 2 males and 14 females, who were in their third trimester of pregnancy. The weight varied between 25 and 30 kg. All the pregnant cows were slaughtered in slaughterhouses under the supervision of the government health inspection service. All procedures were performed on cadavers.

Bovine fetuses were categorized into two groups: the control group (CG, n = 8) treated via umbilical abdominorrhaphy with laparotomy and the videosurgical group (VG, n = 8) treated via video-assisted percutaneous suture technique, with two lateral accesses on the right flank. A wall defect of approximately 15 cm was created to simulate hernia correction, with the VG as a study model. Subsequently, the defect was corrected via percutaneous suture assisted by videolaparoscopy.

Operative techniques

The umbilical abdominorrhaphy in the CG was performed according to the methods described by Sutradhar et al. [12]. The animals were placed in ventral–dorsal recumbency. An elliptical skin incision was made at both ends of the base of the umbilicus using a number 24 scalpel blade (Two Arrows Scalpel Blade, Shanghai Med., SN, China), and the excess was removed for better apposition. The incision was lateralized to the foreskin in male calves. A blunt dissection was performed in the subcutaneous tissue with surgical scissors to create a lesion in the abdominal cavity, which passed through the rectus abdominis muscle to incise the peritoneum.

The umbilical structures were detached from the abdominal wall with partial resection of the vein, umbilical arteries, and urachus using a Miller's knot. All sutures were performed with 0.40-mm nylon thread (Linha de pesca Dourado Premium, Dourado, Londrina, Paraná, Brazil). Umbilical abdominorrhaphy was performed with an interrupted double-breasted suture to correct the defect (Fig 1A–1C). The subcutaneous tissue was sutured with continuous stitches using the Reverdin needle (Fig 1D), and the skin was sutured with interrupted stitches using a U-shaped suture (Fig 1E).

The methodology adopted by Monteiro et al. [9] and Prządka et al. [13] was used for the establishment of access portals, creation of a study model with the defect in the abdominal wall, and the execution of video-assisted percutaneous suturing technique. The access ports were established in the right flank near the paralumbar fossa, caudal to the ribs, with the direct introduction of the trocar via a parietal incision. Skin incisions of approximately 8–10 mm for 10-mm ports and 3–5 mm for 5-mm ports were made using a scalpel to insert the trocars transmurally into the abdominal cavity while maintaining the triangulation of the access doors.

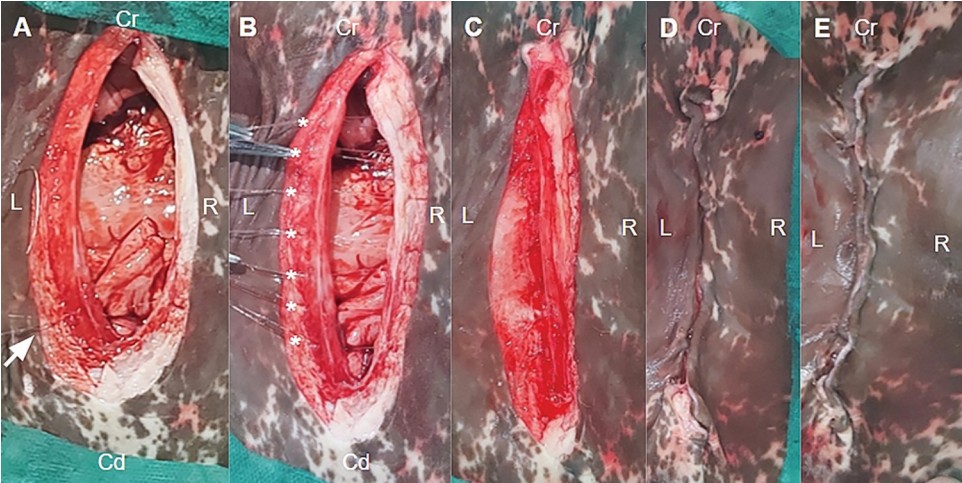

**Fig 1. Umbilical abdominorrhaphy and dermorrhaphy in the laparotomy technique.** (A) Beginning of the suture (white arrow) using interrupted stitches, jacket type. (B) Stitches (*) before being finished. (C) Peritoneum/muscle suture completed. (D) Subcutaneous suture to reduce dead space, and (E) dermorrhaphy with U suture. Cr, cranial; Cd, caudal; L, left; R, right.

A 10-mm rigid endoscope with a diameter of 0˚ (Karl Storz SE & Co, Tuttlingen, Germany) was used, which was coupled to the microcamera system (Combo Endosurgery System, GDI, Ribeirão Preto, São Paulo, Brazil) and lighting cable (Cable of Light 495 Optical Fiber, Karl Storz SE & Co, Germany), connected to the light source (Led Light Source, GDI, Ribeirão Preto, São Paulo, Brazil). Zscan (Image Capture Software, Zscan, Goiânia, Goiás, Brazil) was used for capturing the images. Anatomical specimens of the VG group were placed in the left lateral recumbency position and subjected to laparoscopy using two laparoscopic access ports in the right flank, with one 10-mm cannula in the first and one 5-mm cannula in the second port for access.

A skin incision of approximately 5 mm was made for the second portal, and both the trocars were inserted directly (direct introduction of the trocar through a parietal incision). The assistant operated the optic toward the umbilical base, and the clipping and resection of the umbilical vein, urachus, and umbilical arteries were simulated using laparoscopic scissors, with a cut close to the umbilical ring (UR) (Fig 2A and 2B). The umbilical structures were detached with partial resection, and dissection was performed with laparoscopic scissors in the umbilical base (Fig 2C and 2D), thereby creating a lesion of approximately 15 cm in the abdominal wall (Fig 2E). The opening was confirmed with the visualization on the monitor (Fig 2F).

Video-assisted percutaneous suture was performed after confirmation of the injury in the abdominal wall in the VG. The defect was corrected using a rigid endoscope with internal suturing of the edges of the wound percutaneously. Isolated stitches were applied with the aid of a catheter (16G Teflon Intravenous Peripheral Catheter, Descarpack, São Paulo, São Paulo, Brazil) passing through the nylon thread.

All catheter movements were performed from outside the abdominal cavity, with direct camera control (Fig 3A and 3B). The mandrel of the transcutaneous catheter was introduced at each point, crossing the two edges of the abdominal defect (Figs 4A, 4B, 5A and 5B). The nylon thread was passed through the guide catheter, crossing the edges of the wound (Figs 4C and 5C). The catheter was passed through the same dermal orifice but lateralizing the passage approximately 15 mm in the muscular layer at the edge of the surgical wound. The thread was again passed through the guide catheter, and the surgeon's knot was performed (Fig 4D and 4E). Both the distal and proximal ends of the knot were subcutaneous (Figs 4F and 5D).

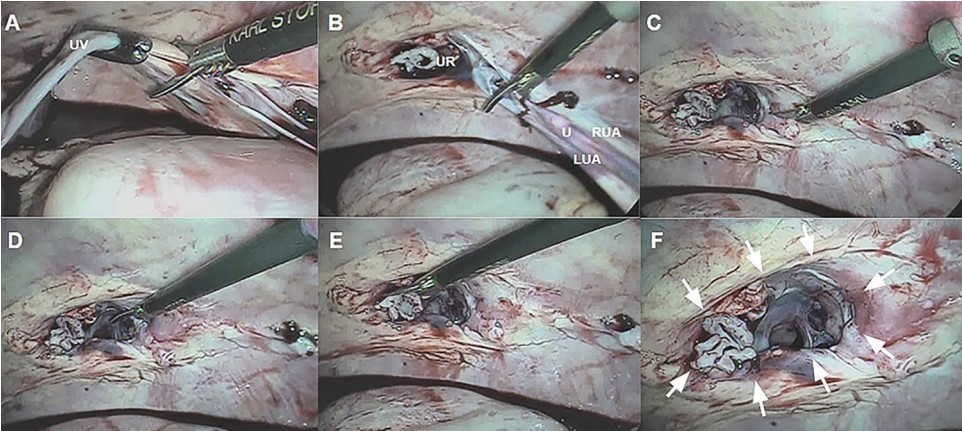

**Fig 2. Creation of the study model for performing the percutaneous suture.** (A) Resection of the umbilical vein (UV). (B) Resection of the right (RUA) and left (LUA) umbilical arteries and urachus (U). (C) Umbilical ring (UR) after resection of the umbilical structures. (D) Beginning of the abdominal wall lesion. (E) End of the abdominal wall lesion. (F) Final result of the defect (white arrows) in the abdominal wall.

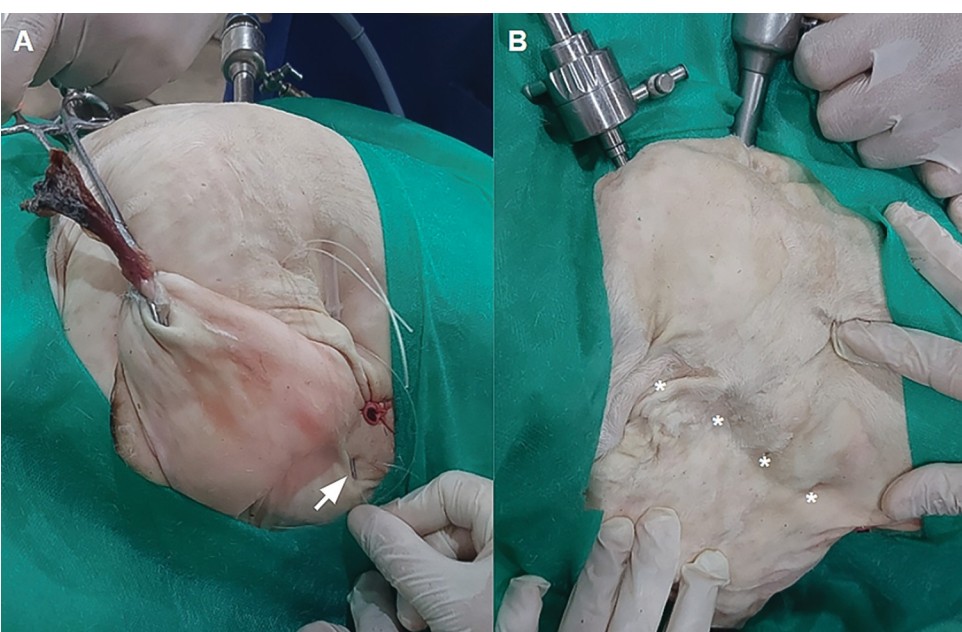

**Fig 3. External view during the video-assisted percutaneous suturing technique.** (A) Beginning of the suture using the guide catheter (white arrow) and (B) completion of the percutaneous suture with isolated stitches (*).

The pneumoperitoneum was undone, the trocar cannulas and laparoscopic portals were removed, and two or three more sutures were performed according to the size of the abdominal wall defect. Myorrhaphy and dermorrhaphy were performed to make the sutures with crossed stitches (Sultan) and U suture in each incision, respectively.

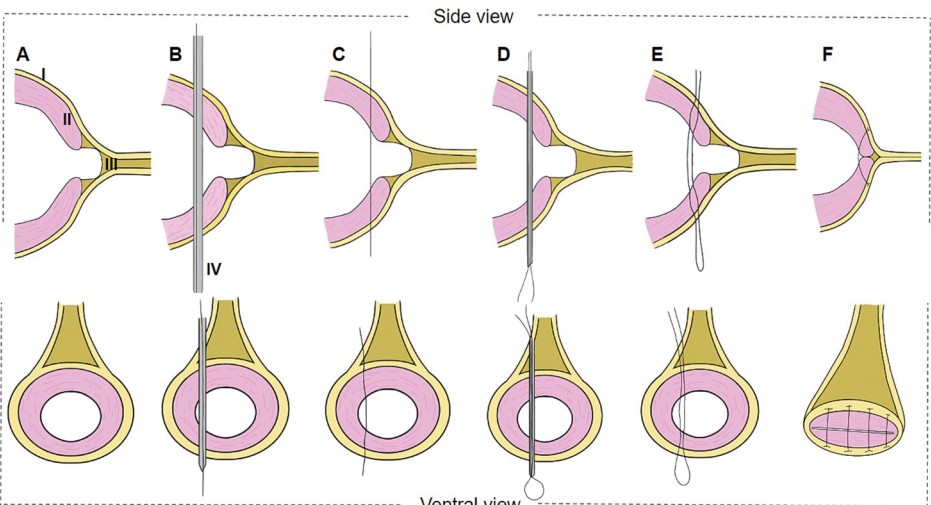

**Fig 4. Schematic representation of the video-assisted percutaneous suturing technique.** (A) Identification of anatomical structures. (B) Introduction of the guiding catheter on one side of the wound edges and placement of the nylon thread. (C) Nylon thread passed through the wound edges after removal of the catheter. (D) Nylon thread re-passed via the catheter directed on the opposite side to the first placement. (E) Nylon thread after removal of the catheter at the edges of the wound. (F) Finishing the stitch with a surgery knot in the subcutaneous region. I-Skin. II-Muscle. III-Subcutaneous. IV-Catheter.

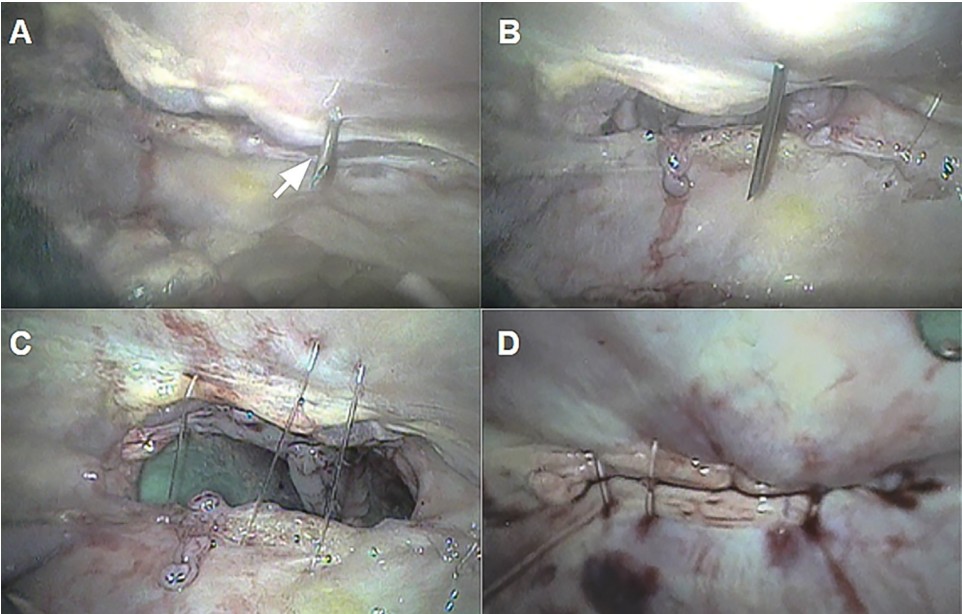

**Fig 5. Internal view during video-assisted percutaneous suturing technique.** (A) Introduction of the mandrel (white arrow) of the catheter into the abdominal cavity, (B) Beginning of the first stitch, with the mandrel directed to the other side of the edge of the surgical wound, (C) Nylon threads applied to the surgical wound, and (D) Completion of the video-assisted percutaneous suture technique.

## Intraoperative analyses

The transoperative time was measured for each stage, and any intercurrence at each stage was recorded in all procedures. Data on total operative time and steps were subjected to descriptive statistics for CG and VG.

The procedures involved the following steps: step 1 (E1): access to visualization and exploration of the umbilical base, resection of the umbilical structures, and creation of the lesion in the abdominal wall, surgical techniques; step 2 (E2): umbilical abdominorrhaphy via laparotomy in the CG group and video-assisted percutaneous suturing of the wound edges in the VG group and final exploration until reversal of the accesses.

## Statistical analysis

The Shapiro–Wilk test was used to confirm the normal distribution of the data. The t-test was used to compare the total operative time of each step. The Mann–Whitney test was used for non-normal distribution of the data. Statistical evaluation was performed using the Bioestat 5.3 package, and $p < 0.05$ was considered significant.

## Results

The bovine fetus model was effective in establishing the operative technique, with visualization, manipulation of the structures, creation of the incision in the abdominal wall, correction with percutaneous suture, and accomplishment of the comparison between the groups. The technique was performed in all 16 animals with visualization of the umbilical structures without major complications. The right lateral approach, with the establishment of laparoscopic accesses, allowed the execution of the procedures properly, as it guaranteed access to the abdominal wall defect with a wide field of view, contributing to the feasibility of the technique.

**Table 1. Surgical time in the different stages.**

| Step | CG (min) | VG (min) | Valor de p |
|---|---|---|---|
| E1 | 5.38 ± 0.23 | 5.73 ± 0.15 | 0.0032 |
| E2 | 33.10 ± 0.43 | 10.13 ± 0.68 | <0.0001* |
| Total surgical time | 38.48 ± 0.35 | 15.86 ± 0.67 | <0.0001* |

Surgical time in the steps of abdominirrhaphy after laparotomy and percutaneous suture guided by laparoscopy in bovine fetuses. CG: Control group, VG: Video-surgical group, p: Probability of significance.

There was no statistical difference between the groups (Table 1) in step 1 with regard to performing the accesses, viewing and exploring the umbilical base, resecting the umbilical structures, and creating the lesion in the abdominal wall. The size of the incisions was 9.4–14.5 cm, with a mean of 11.73 cm. Two incisions of approximately 5–10 mm were made via laparoscopy for the introduction of the two portals.

Resection of the umbilical structures via laparotomy was performed with a Miller's knot. The simulation of clipping of the umbilical vein (UV), urachus, and umbilical arteries was performed via laparoscopy. The defect in the abdominal wall of the umbilical base was established with laparoscopic scissors after video-assisted resection of the umbilical structures.

Surgical techniques were performed, and the final exploration until reversal of the accesses was done in step 2. The creation of the abdominal wall defect was possible with video-assisted percutaneous suture and umbilical abdominorrhaphy via laparotomy. A statistically significant difference was observed between the groups in this aspect (Table 1). There was a small intercurrence during the performance of the suture in the VG group, where the base of the catheter was detached from the mandrel but without compromising the technique.

Diaeresis was performed to close the abdominal cavity with more layers of sutures, and dermorrhaphy was done with broken points in the postsurgical wound of 15–20 cm in the CG group. Dermorrhaphy in the postoperative wound of approximately 5 and 10 mm was performed in the VG group. The total surgical time was longer in the CG group than that in the VG group (Table 1), and stage 2 mostly influenced the duration of the total surgical procedure.

## Discussion

Percutaneous suturing has been used in humans [14], canines [15], and swine [13]. Percutaneous internal ring suturing is a minimally invasive surgical technique for laparoscopic hernia repair. Umbilical hernias are the most frequent form of hernias reported in calves. These hernias can be caused by delayed closure of the UR owing to morphological changes [2], congenital defects, or infections of the umbilical vessels or urachus [16]. The treatment involves surgery performed via laparotomy and other methods [7]. In this study, the proposed technique involved creating a wound in the abdominal wall and using laparoscopy to perform the percutaneous suture.

The umbilical structures, liver, and abomasum were partially visualized in the abdominal cavity during the accesses via laparotomy. The UV and a large part of the hepatic parenchyma were initially visualized via laparoscopy, with great access to a large part of the abdominal cavity. The UV was first inspected as the laparoscope was focused directly on it [9]. The remaining umbilical structures were located by directing the laparoscope toward the umbilical base.

Infected umbilical structures were resected via laparotomy and decrease of the bacterial load to improve prognosis [17]. The resection of the umbilical structures can be en bloc for the surgical treatment of infections of the umbilical vessels, persistent urachus, and umbilical hernias via laparoscopy as well as the conventional technique [18].

The clipping of the internal umbilical structures was simulated with subsequent resection in the VG group. This procedure was performed easily. The positioning of the laparoscopic portals allowed direct access to the structure, thus avoiding perforation of organs or tissues and safe and uncomplicated execution during the procedure [19,20].

Bovine fetuses (corpses) were used in the present study to obtain a study model by creating a defect in the abdominal wall. A lesion was created to simulate the anatomical changes caused by umbilical hernias. This allowed the practice and training of the execution of the technique of umbilical abdominorrhaphy in the CG and the video-assisted percutaneous suture in the VG group. Training is required to perform a laparoscopic technique accurately. Some study models that help surgeons acquire basic laparoscopic skills exist [21].

Umbilical hernias are ovoid defects in the ventral abdominal wall [6,17]. These defects can be repaired using two surgical methods: closed and open. Simple hernias measuring 1–3 cm are common, but they can also be >3 cm with the presence of organs or with serious complications [1]. Umbilical herniorrhaphy is recommended for hernias >5 cm. This procedure in bovine fetuses was performed during umbilical herniorrhaphy after the incision of the hernia sac and peritoneum, with the introduction of the viscera into the abdominal cavity [2,20].

Abdominal wall injuries created in the tested groups were adequate for performing the techniques. A 16-G catheter was used to make the percutaneous suture with the aid of an injection needle to perform the extracorporeal suture in the VG group [14,15]. This technique is a minimally invasive alternative that is easy to perform [13]. Laparoscopic techniques to reduce hernias have a faster recovery after the procedures compared with conventional methods [7].

The total surgical time was statistically longer in the CG group than that in the VG group. The largest incisions via laparotomy and the suturing of the subcutaneous tissue with simple anchorage using continuous stitches in the musculature, in addition to dermorrhaphy, were performed with pattern interrupted suture [16]. The time for correction of the abdominal wall defect in the umbilical region was shorter with the aid of laparoscopy [7,9]. This suture has lower complication rates than conventional surgical techniques in video-assisted surgery [7,22].

## Conclusions

The study model allowed the execution of the video-assisted percutaneous suture technique for wound repair in the abdominal wall with less surgical time compared with the conventional technique. However, further studies are needed in a larger cohort, including live animals that require surgical procedures for umbilical hernias to confirm the results of the present study.

## Supporting information

**S1 File.**
(DOCX)

## Acknowledgments

**Ethics committee approval:** This research was approved by the Ethics Committee on the Use of Animals at the Universidade de Federal do Pará (CEUA/UFPA N°. 4848261017).

## Author Contributions

**Conceptualization:** Carla Rozilene Guimarães Silva, Pedro Paulo Maia Teixeira.

**Data curation:** Carla Rozilene Guimarães Silva, Thiago da Silva Cardoso, Késia Bandeira da Silva, Heytor Jales Gurgel, João Pedro Monteiro Barroso, Luiz Henrique Vilela Araújo, Luis Enrique Soza Altamirano, Loise Araújo de Sousa, Luiza Paula Araújo Alcântara, Marcos Emanoel Martins Ferreira, Lucas Santos Carvalho, José Leandro da Silva Gonçalves, Jhoisse Hamar Guimarães Rodrigues, Francisco Décio de Oliveira Monteiro, Rinaldo Batista Viana, Pedro Paulo Maia Teixeira.

**Formal analysis:** Carla Rozilene Guimarães Silva.

**Investigation:** Heytor Jales Gurgel, João Pedro Monteiro Barroso, Francisco Décio de Oliveira Monteiro, Rinaldo Batista Viana, Pedro Paulo Maia Teixeira.

**Methodology:** Heytor Jales Gurgel, João Pedro Monteiro Barroso, Pedro Paulo Maia Teixeira.

**Project administration:** Pedro Paulo Maia Teixeira.

**Resources:** Francisco Décio de Oliveira Monteiro, Rinaldo Batista Viana, Pedro Paulo Maia Teixeira.

**Supervision:** Francisco Décio de Oliveira Monteiro, Rinaldo Batista Viana, Pedro Paulo Maia Teixeira.

**Validation:** Heytor Jales Gurgel, João Pedro Monteiro Barroso, Francisco Décio de Oliveira Monteiro, Rinaldo Batista Viana, Pedro Paulo Maia Teixeira.

**Visualization:** Thiago da Silva Cardoso, Késia Bandeira da Silva, Heytor Jales Gurgel, João Pedro Monteiro Barroso, Luiz Henrique Vilela Araújo, Luis Enrique Soza Altamirano, Loise Araújo de Sousa, Luiza Paula Araújo Alcântara, Marcos Emanoel Martins Ferreira, Lucas Santos Carvalho, José Leandro da Silva Gonçalves, Jhoisse Hamar Guimarães Rodrigues, Rinaldo Batista Viana, Pedro Paulo Maia Teixeira.

**Writing – original draft:** Carla Rozilene Guimarães Silva, Thiago da Silva Cardoso, Késia Bandeira da Silva, Heytor Jales Gurgel, João Pedro Monteiro Barroso, Luiz Henrique Vilela Araújo, Luis Enrique Soza Altamirano, Loise Araújo de Sousa, Luiza Paula Araújo Alcântara, Marcos Emanoel Martins Ferreira, Lucas Santos Carvalho, José Leandro da Silva Gonçalves, Jhoisse Hamar Guimarães Rodrigues, Francisco Décio de Oliveira Monteiro, Rinaldo Batista Viana, Pedro Paulo Maia Teixeira.

**Writing – review & editing:** Thiago da Silva Cardoso, Késia Bandeira da Silva, Heytor Jales Gurgel, João Pedro Monteiro Barroso, Luiz Henrique Vilela Araújo, Luis Enrique Soza Altamirano, Loise Araújo de Sousa, Luiza Paula Araújo Alcântara, Marcos Emanoel Martins Ferreira, Lucas Santos Carvalho, José Leandro da Silva Gonçalves, Jhoisse Hamar Guimarães Rodrigues, Francisco Décio de Oliveira Monteiro, Rinaldo Batista Viana, Pedro Paulo Maia Teixeira.

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
