## [Decision Letter · Decision Letter 0]

21 Nov 2022

PONE-D-22-23559Laparoscopy-assisted percutaneous correction of abdominal wall defects in the umbilical region in an animal modelPLOS ONE

Dear Dr. Silva,

Thank you for submitting your manuscript to PLOS ONE. After careful consideration, we feel that it has merit but does not fully meet PLOS ONE’s publication criteria as it currently stands. Therefore, we invite you to submit a revised version of the manuscript that addresses the points raised during the review process.

Reviewers raised minor concerns about the manuscript. It is recommended that the authors follow the suggestions of the reviewers and then modify the manuscript accordingly.

In addition, the following aspects need to be clarified or changed. 

English-language proofreading of the entire text of the manuscript is required (please have the correct form of writing by a native English speaker or use an English writing service); 

More specifics about the method used are required, especially in the VG group, so that the reader will potentially be able to reproduce it, in particular:

-precisely describe the positioning of the trocars and ports (it is not enough to write "through the right flank");

-methods of entry of the CO2 into the abdominal cavity in the induction phase of the pneumoperitoneum (for example use of the Verres needle or direct introduction of the trocar through a parietal incision?);

-please specify the length of the catheter used.

In Statistical analysis, what’s the meaning of “with the post test”? Perhaps do authors mean a post-hoc test? But if so, it seems inappropriate given that, being a comparison between two groups, a t-test was rightly done, and not an ANOVA. Furthermore, it is not necessary to specify "Wilcoxon" in parentheses, “Mann-Whitney test” is sufficient. 

In abstract and method section the authors define the surgical steps with S1 and S2, then in results (Table 1) with E1 and E2: please homogenize by using the same acronyms in the whole manuscript (the same for GV and VG, as suggested by the reviewer). 

Authors should please indicate major limitations of the study. 

We look forward to receiving your revised manuscript.

Kind regards,

Adolfo Maria Tambella, DVM, MSc

Academic Editor

PLOS ONE

Journal Requirements:

“The authors would like to thank CAPES and PROPESP/UFPA for financially supporting this study.”

“The authors would like to thank CAPES and PROPESP/UFPA for financially supporting this study.”

“The authors would like to thank CAPES and PROPESP/UFPA for financially supporting this study.”

4. Please expand the acronym “CAPES and PROPESP/UFPA” (as indicated in your financial disclosure) so that it states the name of your funders in full.

This information should be included in your cover letter; we will change the online submission form on your behalf

“The authors declare that they have no known competing financial interests.”

7. In your Data Availability statement, you have not specified where the minimal data set underlying the results described in your manuscript can be found. PLOS defines a study's minimal data set as the underlying data used to reach the conclusions drawn in the manuscript and any additional data required to replicate the reported study findings in their entirety. All PLOS journals require that the minimal data set be made fully available. For more information about our data policy, please see http://journals.plos.org/plosone/s/data-availability.

Reviewers' comments:

Reviewer's Responses to Questions

**Comments to the Author**

1. Is the manuscript technically sound, and do the data support the conclusions?

Reviewer #1: Yes

Reviewer #2: Yes

Reviewer #3: Yes

2. Has the statistical analysis been performed appropriately and rigorously? 

Reviewer #1: Yes

Reviewer #2: Yes

Reviewer #3: Yes

3. Have the authors made all data underlying the findings in their manuscript fully available?

Reviewer #1: Yes

Reviewer #2: Yes

Reviewer #3: Yes

4. Is the manuscript presented in an intelligible fashion and written in standard English?

Reviewer #1: Yes

Reviewer #2: Yes

Reviewer #3: Yes

5. Review Comments to the Author

Reviewer #1: The study was done to develop a study model for laparoscopic surgical correction of umbilical defects in calves. The manuscript is well presented and can be accepted with minor corrections mentioned below.

Title: The title should include "study model" as no live animal was involved in the surgical correction to observe the prognosis.

Abstract: The video-surgical group (line 33) was termed VG but in Materials and methods it was denoted GV (lines 87, 89, 141). Please correct this.

Line 109: the meaning of Cd is should be caudal.

In fig 4, please mention the labeling of I-IV.

Statistical analysis: p < 0.05 should be considered significant.

Reviewer #2: It is interesting work with great finding.

However, I have some comments that should be addressed to increase clarity

Abstract:

Add a good background and conclusive statement for the abstract

Introduction:

Better to indicate the differential dx for hernia in general and umbilical hernia in specific such as hematoma, tumor....

Material and methods:

Better to say Study animals/ experimental animals

What breed of cattle were used?

Regarding your criteria for inclusion, why do you like to use pregnant animals as your experimental animal? Do you think the case more frequent on pregnant animals if so please indicate this on the introduction section.

Please indicate limitation of you study.

Discussion

Please try to also provide an update on what new technique can be adopted from this experimental study as compared to already existing techniques? This will increase visibility of your research.

Conclusion

It should be related with your main findings of the experiment.

Reviewer #3: Accepted without any comments, In fact, this study contains a scientific addition and a new and modern method in laparoscopy without problems or complications, and the care after the operation is simple and uncomplicated.

6. PLOS authors have the option to publish the peer review history of their article (what does this mean?). If published, this will include your full peer review and any attached files.

Reviewer #1: **Yes: **Moinul Hasan

Reviewer #2: **Yes: **Haben Fesseha

Reviewer #3: No

---

## [Author Response · Author response to Decision Letter 0]

27 Feb 2023

Academic Editor

English-language proofreading of the entire text of the manuscript is required (please have the correct form of writing by a native English speaker or use an English writing service.

Response:

More specifics about the method used are required, especially in the VG group, so that the reader will potentially be able to reproduce it, in particular:

-precisely describe the positioning of the trocars and ports (it is not enough to write "through the right flank");

Response: Suggestion inserted in the manuscript (113-117)

-methods of entry of the CO2 into the abdominal cavity in the induction phase of the pneumoperitoneum (for example use of the Verres needle or direct introduction of the trocar through a parietal incision?);

Response: Suggestion inserted in the manuscript (127)

-please specify the length of the catheter used.

Response: Suggestion inserted in the manuscript (124)

In Statistical analysis, what’s the meaning of “with the post test”? Perhaps do authors mean a post-hoc test? But if so, it seems inappropriate given that, being a comparison between two groups, a t-test was rightly done, and not an ANOVA. Furthermore, it is not necessary to specify "Wilcoxon" in parentheses, “Mann-Whitney test” is sufficient.

Response: Corrected in manuscript (186)

In abstract and method section the authors define the surgical steps with S1 and S2, then in results (Table 1) with E1 and E2: please homogenize by using the same acronyms in the whole manuscript (the same for GV and VG, as suggested by the reviewer). 

Response: Corrected in manuscript

Authors should please indicate major limitations of the study

As limitations of this study, the laparoscopic techniques are influenced by the surgeon's experience, in cases of very large hernias they can be difficult, not using meshes in these procedures and obtaining bovine fetuses to perform the techniques.

Reviewer Comments:

5. Review Comments to the Author

Reviewer 1: 

The study was done to develop a study model for laparoscopic surgical correction of umbilical defects in calves. The manuscript is well presented and can be accepted with minor corrections mentioned below.

Title: The title should include "study model" as no live animal was involved in the surgical correction to observe the prognosis.

Response: Suggestion included in the manuscript (2)

Abstract: The video-surgical group (line 33) was termed VG but in Materials and methods it was denoted GV (lines 87, 89, 141). Please correct this.

Response: Corrected in manuscript

Line 109: the meaning of Cd is should be caudal.

Response: Corrected in manuscript

In fig 4, please mention the labeling of I-IV.

Response: Corrected in manuscript (163)

Statistical analysis: p < 0.05 should be considered significant.

Response: Corrected in manuscript (187-188)

Reviewer 2: It is interesting work with great finding.

However, I have some comments that should be addressed to increase clarity

Abstract:

Add a good background and conclusive statement for the abstract

Response: Suggestion included in the manuscript

Introduction:

Better to indicate the differential dx for hernia in general and umbilical hernia in specific such as hematoma, tumor....

Response: Suggestion included in the manuscript

Material and methods:

Better to say Study animals/ experimental animals

What breed of cattle were used?

Response: Mixed breed animals.

Regarding your criteria for inclusion, why do you like to use pregnant animals as your experimental animal? Do you think the case more frequent on pregnant animals if so please indicate this on the introduction section.

Response: We did not use pregnant animals in the study, we used fetuses obtained from pregnant animals that were slaughtered for consumption.

We use dead fetuses as these are newly developed techniques and surgeons need to gain necessary skills in cadavers without causing pain or suffering to live animals.

Please indicate limitation of you study.

The limitations of this study were during the collection and conservation of bovine fetuses. Most of the cows that were destined for slaughter were not pregnant and the slaughterhouse was far from the place where this experiment was carried out. To get the total number of bovine fetuses planned, several displacements were necessary.

After obtaining the fetuses, they were kept under refrigeration for conservation until the period of the experiment. In the first moment, days before the practices, there was a power outage, the refrigerated chamber stopped working and all the animals were not suitable for use, they were lost. In the second moment after obtaining more animals, the experiment was carried out.

Discussion

Please try to also provide an update on what new technique can be adopted from this experimental study as compared to already existing techniques? This will increase visibility of your research.

Conclusion

It should be related with your main findings of the experiment.

---

## [Decision Letter · Decision Letter 1]

27 Mar 2023

PONE-D-22-23559R1Laparoscopy-assisted percutaneous correction of abdominal wall defects in the umbilical region in an animal model (study model)PLOS ONE

Dear Dr. Silva,

Thank you for submitting your manuscript to PLOS ONE. After careful consideration, we feel that it has merit but does not fully meet PLOS ONE’s publication criteria as it currently stands. Therefore, we invite you to submit a revised version of the manuscript that addresses the points raised during the review process. The quality of the manuscript has improved since the first revision, however there are still minor revisions to be made. 

English-language proofreading of the new version of the whole text is still suggested to increase the clarity of the manuscript.

The following specific aspects need to be clarified or changed.

-Keywords: please replace “invasive minimally technique” with “minimally invasive technique”.

-Title: please avoid redundancy in the title; First suggested option for title: “Laparoscopy-assisted percutaneous correction of abdominal wall defects in the umbilical region in an animal model”. Second suggested option for title: “Laparoscopy-assisted percutaneous correction of abdominal wall defects in the umbilical region in a cadaveric model of bovine fetus”.

-lines 37-42: please rephrase to facilitate the reader's understanding.

-line 89: please replace “All these pregnant cows” with “All the pregnant cows”.

-line 91: delete “without causing pain or suffering to the animals.” To avoid redundancy as cadavers are used.

-line 126: please replace "lighting" with "lighting cable".

-lines 164-169 (capture of figure 4): please replace “nylon” with “nylon thread”, as in caption of figure 5. 

-line 187: please replace “, surgical techniques:” with “;”

-lines 202-203: please rephrase to facilitate the reader's understanding. 

-line 212: Since in the table is reported the p-values, it is suggested to delete “p≤0.05” from the caption.

-line 222: It is not clear what these values “(08/01–12.5%)” indicates. Maybe you mean “(1 case out of 8, 12.5%)”? Please clarify or delete. 

-lines 224-225: please rephrase to facilitate the reader's understanding.

-line 236: please delete “and other methods”.

-lines 271-273: please rephrase to facilitate the reader's understanding.

We look forward to receiving your revised manuscript.

Kind regards,

Adolfo Maria Tambella, DVM, MSc

Academic Editor

PLOS ONE

Journal Requirements:

Reviewers' comments:

Reviewer's Responses to Questions

**Comments to the Author**

1. If the authors have adequately addressed your comments raised in a previous round of review and you feel that this manuscript is now acceptable for publication, you may indicate that here to bypass the “Comments to the Author” section, enter your conflict of interest statement in the “Confidential to Editor” section, and submit your "Accept" recommendation.

Reviewer #1: All comments have been addressed

2. Is the manuscript technically sound, and do the data support the conclusions?

Reviewer #1: Yes

3. Has the statistical analysis been performed appropriately and rigorously? 

Reviewer #1: Yes

4. Have the authors made all data underlying the findings in their manuscript fully available?

Reviewer #1: Yes

5. Is the manuscript presented in an intelligible fashion and written in standard English?

Reviewer #1: Yes

6. Review Comments to the Author

Reviewer #1: (No Response)

7. PLOS authors have the option to publish the peer review history of their article (what does this mean?). If published, this will include your full peer review and any attached files.

Reviewer #1: **Yes: **Moinul Hasan

---

## [Author Response · Author response to Decision Letter 1]

29 Apr 2023

Ref: PONE-D-22-23559R1

Laparoscopy-assisted percutaneous correction of abdominal wall defects in the umbilical region in an animal model (study model)

Answers to the reviewers

Academic Editor

English-language proofreading of the new version.

Response: Yes (Enclosed English proofreading certificate).

- Keywords: please replace “invasive minimally technique” with “minimally invasive technique”

Response: Suggestion inserted in the manuscript.

-Title: please avoid redundancy in the title; First suggested option for title: “Laparoscopy-assisted percutaneous correction of abdominal wall defects in the umbilical region in an animal model”. Second suggested option for title: “Laparoscopy-assisted percutaneous correction of abdominal wall defects in the umbilical region in a cadaveric model of bovine fetus”.

Response: Second suggestion inserted in the manuscript.

-lines 37-42: please rephrase to facilitate the reader's understanding.

Response: Suggestion inserted in the manuscript.

“The procedures were performed in two steps. The first step consisted of creating an abdominal wall defect in the umbilical region by laparoscopic approach in an iatrogenic manner (Step 1: E1). The second stage consisted of conventional abdominorrhaphy of the umbilical region wall defect in the CG group and video-assisted percutaneous suturing of the edges of the iatrogenic abdominal wall defect in the VG group, until reversal of the laparoscopic accesses (Step 2: E2).”

-line 89: please replace “All these pregnant cows” with “All the pregnant cows”.

Response: Suggestion in manuscript.

-line 91: delete “without causing pain or suffering to the animals.” To avoid redundancy as cadavers are used. 

Response: Suggestion inserted in manuscript.

-line 126: please replace "lighting" with "lighting cable".

Response: Correction inserted in manuscript.

-lines 164-169 (capture of figure 4): please replace “nylon” with “nylon thread”, as in caption of figure 5.

Response: Suggestion inserted in manuscript.

-line 187: please replace “, surgical techniques:” with “;”

Response: Substitution realized in manuscript.

-lines 202-203: please rephrase to facilitate the reader's understanding.

Response: Correction inserted in manuscript.

“The right lateral approach, with the establishment of laparoscopic accesses, allowed the execution of the procedures properly, as it guaranteed access to the abdominal wall defect with a wide field of view, contributing to the feasibility of the technique.”

-line 212: Since in the table is reported the p-values, it is suggested to delete “p≤0.05” from the caption.

Response: Suggestion inserted in manuscript.

-line 222: It is not clear what these values “(08/01–12.5%)” indicates. Maybe you mean “(1 case out of 8, 12.5%)”? Please clarify or delete.

Response: Values excluded in the manuscript.

-lines 224-225: please rephrase to facilitate the reader's understanding.

Response: Correction inserted in manuscript.

“Dieresis of the musculature and peritoneum was performed to correct the iatrogenic defect of the abdominal wall with subsequent dermarrhaphy by separate simple suture in the CG group.”

-line 236: please delete “and other methods”.

Response: Suggestion inserted in manuscript.

-lines 271-273: please rephrase to facilitate the reader's understanding.

Response: Correction inserted in manuscript.

“The time for correction of the abdominal wall defect in the umbilical region was shorter with the aid of laparoscopy”.

---

## [Editor Report · Decision Letter 2]

7 May 2023

Laparoscopy-assisted percutaneous correction of abdominal wall defects in the umbilical region in an animal model (study model)

PONE-D-22-23559R2

Dear Dr. Silva,

We’re pleased to inform you that your manuscript has been judged scientifically suitable for publication and will be formally accepted for publication once it meets all outstanding technical requirements.

Kind regards,

Adolfo Maria Tambella, DVM, MSc

Academic Editor

PLOS ONE

Additional Editor Comments (optional):

After carefully considering the old and new versions of the manuscript, the scientific quality has grown and now it can be considered suitable for publication. Congratulations to the authors.
---

## [Editor Report · Acceptance letter]

22 May 2023

PONE-D-22-23559R2 

*Laparoscopy-assisted percutaneous correction of abdominal wall defects in the umbilical region in a cadaveric model of bovine fetus*

Dear Dr. Silva:

I'm pleased to inform you that your manuscript has been deemed suitable for publication in PLOS ONE. Congratulations! Your manuscript is now with our production department. 

Kind regards, 

on behalf of

Dr. Adolfo Maria Tambella 

Academic Editor

PLOS ONE